## How valid are projections of the future prevalence of diabetes? Rapid reviews of prevalence-based and Markov chain models and comparisons of different models' projections for England

Gwyn Bevan [1], Chiara De Poli [1], Mi Jun Keng [1], Rosalind Raine [2]

¹Department of Management, London School of Economics and Political Science, London, UK
²Department of Applied Health Research, University College London, London, UK

**Correspondence to**
Professor Gwyn Bevan;
r.g.bevan@lse.ac.uk

## ABSTRACT

**Objectives** To examine validity of prevalence-based models giving projections of prevalence of diabetes in adults, in England and the UK, and of Markov chain models giving estimates of economic impacts of interventions to prevent type 2 diabetes (T2D).

**Methods** Rapid reviews of both types of models. Estimation of the future prevalence of T2D in England by Markov chain models; and from the trend in the prevalence of diabetes, as reported in the Quality and Outcomes Framework (QOF), estimated by ordinary least squares regression analysis.

**Setting** Adult population in England and UK.

**Main outcome measure** Prevalence of T2D in England and UK in 2025.

**Results** The prevalence-based models reviewed use sample estimates of past prevalence rates by age and sex and projected population changes. Three most recent models, including that of Public Health England (PHE), neither take account of increases in obesity, nor report Confidence Intervals (CIs). The Markov chain models reviewed use transition probabilities between states of risk and death, estimated from various sources. None of their accounts give the full matrix of transition probabilities, and only a minority report tests of validation. Their primary focus is on estimating the ratio of costs to benefits of preventive interventions in those with hyperglycaemia, only one reported estimates of those developing T2D in the absence of a preventive intervention in the general population.

Projections of the prevalence of T2D in England in 2025 were (in millions) by PHE, 3.95; from the QOF trend, 4.91 and by two Markov chain models, based on our review, 5.64 and 9.07.

**Conclusions** To inform national policies on preventing T2D, governments need validated models, designed to use available data, which estimate the scale of incidence of T2D and survival in the general population, with and without preventive interventions.

## INTRODUCTION

Rigorous analysis of worldwide trends of increases in the preventable onset of type 2 diabetes (T2D) in adults justifies a call for

### Strengths and limitations of this study

► We undertook rapid reviews of prevalence-based models and Markov chain models, which have been used to give projections of the future prevalence of diabetes to examine their data sources and assumptions.

► We compared projections of the future prevalence of diabetes in England from: reports for the prevalence-based models; our own Markov chain models (based on transition probabilities from our review) and the trend in the prevalence of diagnosed diabetes as reported by general practitioners in England (estimated by ordinary least squares regression analysis).

► This study's limitations are that our reviews were rapid and our models are transparent and simple.

the urgent of implementation of 'population-based interventions that prevent diabetes, enhance its early detection and use lifestyle and pharmacological interventions to prevent or delay its progression to complications'.[1] In March 2015, National Health Service (NHS) England and Public Health England (PHE) launched, at scale, the NHS Diabetes Prevention Programme (NDPP), which is a pragmatic lifestyle intervention that targets adults with raised levels of glycated haemoglobin (HbA1c) or a fasting plasma glucose (FPG).[2] The NDPP aims 'to significantly reduce the 4 million people in England otherwise expected to have T2D by 2025' based on evidence from 'well-designed randomised controlled trials in Finland, the USA, Japan, China and India'.[3] Many studies have used Markov chain models to estimate the impacts of such preventive interventions using transition probabilities between states: 'normoglycaemia' and 'intermediate hyperglycaemia (IH)' (glucose levels associated with a low and high risks of developing T2D), T2D and

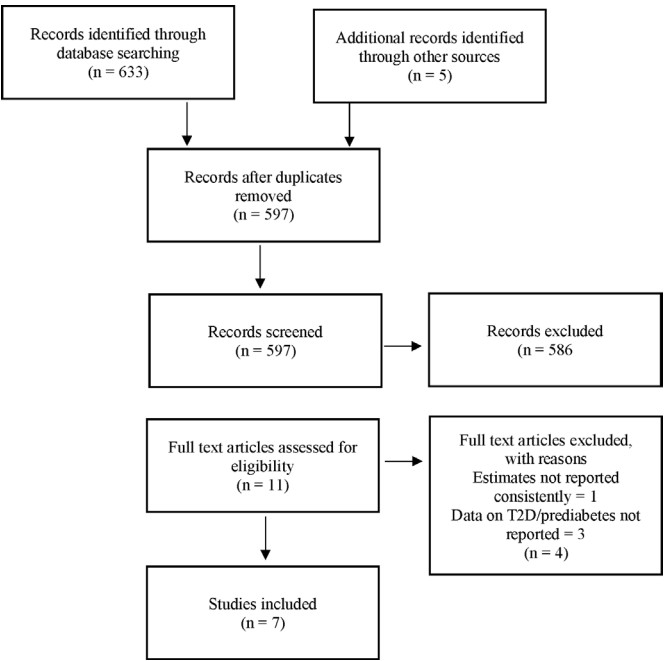

**Figure 1** Review flow chart of epidemiological models. T2D, type 2 diabetes.

death. When we tried to use these models,[4] we had difficulty in finding details from published models, and the models we did develop gave projections of the future prevalence of T2D in 2025 in England, in the absence of a preventive intervention, which were much higher than 4 million. That estimate is based on PHE's prevalence-based model[5] that gives future projections of the prevalence of T2D (at future time t, $N(t)$) by multiplying projections of the country's population by age and sex (at time t $(P(t))$ by projections of age-specific prevalence of diabetes (at

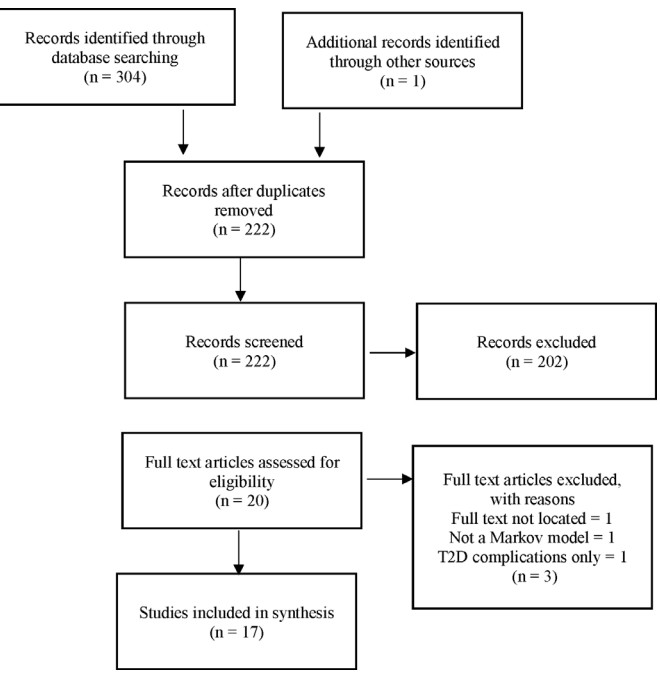

**Figure 2** Review flow charts of Markov chain models. T2D, type 2 diabetes.

time t, $D(t)$). ($N(t)=D(t)*P(t)$).) Hence this study, which had three aims. First, to compare the model used by PHE to project the prevalence of diabetes in England with other models applied to England and the UK. Second, to identify Markov chain models, we could use to project the prevalence of T2D in England. Third, to compare projections for England of prevalence of diabetes and T2D from different models.

Although we have used England for the purpose of comparing projections by these different models, our study raises general questions about their validity. And hence of the evidence available to governments assessing the urgency of preventing T2D and choosing between different interventions. We consider only adults with diabetes. We use 'diabetes' to cover all types of diabetes, T2D for adults with type 2, 'true' prevalence for both diagnosed and undiagnosed diabetes and T2D.

## METHODS
### Rapid reviews
Our comparisons of projections of different models builds on two reviews of the literature, which were designed to be rapid (not systematic): 'a type of knowledge synthesis in which components of the systematic review process are simplified or omitted to produce information in a short period of time'.[6] We used stringent criteria to identify the principal methods of each type of model. These reviews were undertaken in March 2018, of articles published at any time available on Web of Science and PubMed, which together provide a comprehensive coverage of the literature in the medical and applied health research fields. (The search strategy of each review is given in online supplementary appendix 1.) Articles included in each review were critically appraised and technical specifications of the models and projections were extracted and tabulated. The flow charts in figures 1 and 2 show the screening process.

Rapid review 1 aimed to identify primary studies published from 2010 of models giving estimates of the prevalence of diabetes in adults in England or the UK. We examined how the models take account of future changes in age-specific prevalence rates and test their validity.

Rapid review 2 aimed to identify primary studies using Markov chain models that reported results of interventions to prevent T2D. We reviewed articles using Markov models to run economic analyses, utility analyses and cost-effectiveness analyses of preventive interventions including people diagnosed with IH according to different measures: HbA1c, FPG, Impaired Fasting Glucose (IFG) and Impaired Glucose Tolerance (IGT). (Definitions are given in online supplementary appendix 1.) We reviewed the transition probabilities of the different models, and whether they were used to estimate the future prevalence of T2D without a preventive intervention and tests of validation. In our discussion, we refer to the systematic review by Leal *et al*[7] of models of pre-diabetes populations used

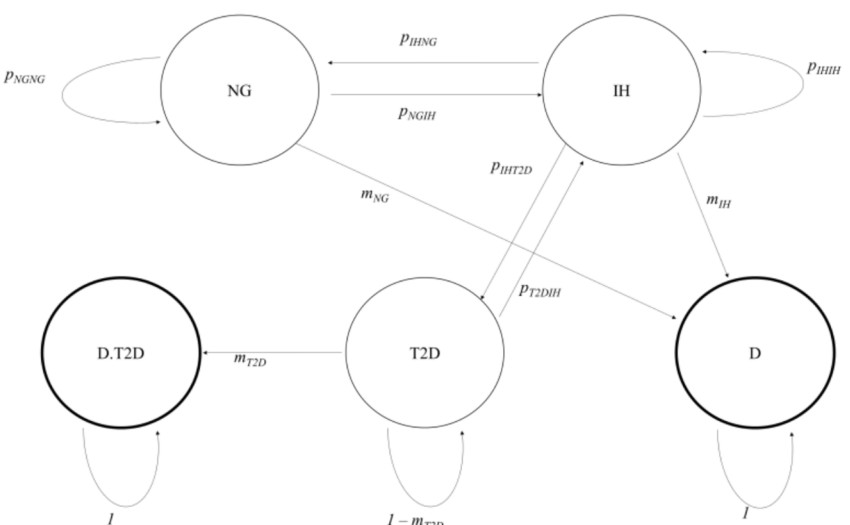

**Figure 3** Our Markov chain model.

### Our Markov chain models
Our Markov chain models are in Excel (see figure 3) and based on a cycle length of 1 year. The transition probabilities between states other than death are based on rapid review 2. We estimated English mortality rates using the following data sources: age distributions for those with IH and diabetes, from combined Health Surveys for England (HSE) data (from 2009 to 2013)[8]; mortality rates by age, from the Office of National Statistics (for 2015)[9]; Hazard Ratios (HRs) for those with diabetes (1.32) and T2D (1.28) with reference to those without diabetes, from the National Diabetes Audit (for 2015–2016).[10] We estimated mortality rates for those with IH using HRs with reference to those with normoglycaemia as estimated (with 95% CIs) by a systematic review and meta-analysis[11]: for IGT 1.32 (1.23 to 1.40) and for HbA1c 0.97 (0.88 to 1.07). We used 1.32 for IGT, but 1 for HbA1c because the estimate of 0.97 is not significantly different from 1. We estimated mortality rates as follows, for 2015, for the English population: for normoglycaemia, 0.6% (compared with 0.9% for the general adult population); for IH, 1.9% and 2.3% for HbA1c and IGT; and for T2D, 2.3% and 2.2% for HbA1c and IGT. The probability of remaining in a state was derived as the residual (so all transition probabilities from each state sum to one).

In making future projections of the prevalence of T2D in England, without a preventive intervention, up to 2035, we used PHE estimates for 2015 of those with diabetes[12] and IH,[13] and derived the estimate of those with normo-glycaemia as the residual for the population of England.[14] Given doubts over the reliability of diagnosing IH,[15] we examined the robustness of our results by using the PHE estimate (IH=5.05 million), and the extreme value of 0 (IH=0). The data sources of our estimates for England, of the prevalence of diabetes, IH and normoglycaemia; and

of mortality rates of those with T2D, IH and normogly-caemia are given in the text.

### Estimating the trend in diagnosed diabetes
We estimated, by OLS regression analysis (using R),[16] the trend increase in the reported prevalence of diabetes as diagnosed by general practitioners in England, in the Quality Outcomes Framework (QOF) from 2004–2005 (2004) to 2017–2018 (2017)).[17] We used these estimates to give projections of the future prevalence of diagnosed diabetes to 2035.

### Comparing projections of the prevalence of diabetes
We compared three sets of projections of the prevalence of diabetes and T2D in England from:
▶ Different prevalence-based models.
▶ The trend in QOF data.
▶ Our Markov chain models.

The ratios we used for making comparisons across different estimates and their sources are as follows:
▶ 75% for the ratio of diagnosed to the true prevalence of diabetes.[18 19]
▶ 90% for the ratio of the prevalence of T2D to diabetes.[12]

### Patients and public involvement
Patients and the public were not involved in this research study.

## RESULTS
### Rapid review 1: methods of prevalence-based models
Rapid review 1 of methods of prevalence-based models retrieved 633 articles and from their citations we identified a further five by snowballing.[20] After removing duplicates, we screened 597 articles, of which 11 were relevant and fully assessed. After reviewing the full articles, five were excluded and seven were included in our analysis.[5 18 21–25] This review identified four different underlying models described in

**Table 1** Methods of prevalence-based models

| Model | Method of estimation | Prevalence rates used for projections | Validation against QOF data? | Model validation? | CIs? |
|---|---|---|---|---|---|
| Shaw et al[21] | Logistic regression | Age and sex | No | No | No |
| Guariguata et al[26] | Logistic regression | Age and sex, and urban/rural | No | No | No |
| Association of Public Health Observatories[18 25] | Direct estimation from HSE for age, sex and IMD. Trend in obesity estimated by linear regression. | Age and sex, IMD (2004), Ethnicity and increases in obesity | Yes for 2008/2009 | No | Yes |
| PHE[5] | Logistic regression | Age and sex, ethnicity, IMD 2015 | Yes for 2014/2015 | Yes: refitting model on 70% of data and assessing against remaining 30% | No |

HSE, Health Surveys for England; IMD, Index of Multiple Deprivation; PHE, Public Health England; QOF, Quality and Outcomes Framework.

table 1, which have been used to give five different projections of the future prevalence of diabetes for England and the UK. Two models produce global estimates: Shaw et al,[21] Guariguata et al,[26] which is used by Whiting et al[22] and Guariguata et al;[23] and two for England only, the PHE model,[5] and the Association of Public Health Observatories (APHO) Diabetes Prevalence Model,[18] which is used by Hex et al[24] and Gatineau et al.[25]

Each prevalence-based model uses: projected population changes; and estimates of the true age-specific prevalence rates of diabetes, from past annual HSE, which are subject to two limitations. First, the small size of the sample means that the point estimate for the year of the survey is surrounded by large CI estimates. Gatineau et al indicate that the HSE survey for 2013 gives point estimate of prevalence of 7.3% with CI estimates ranging from 4.3% to 10.3%.[25] The PHE model[5] reduces the sampling error from HSE by using 3 years of data (2012, 2013 and 2014). Second, the HSE estimates of prevalence are based on those who self-reported a diabetes diagnosis made by a doctor (by HbA1c or FPG); and, for those who have not been diagnosed and agreed to have a blood test, having a HbA1c value of 6.5% or more.[5] Hence these estimates may be in error because of poor reliability of self-reporting or because of actual diagnostic errors. Barry et al (p. 9) report that 'The most commonly used test (HbA1c) is neither sensitive nor specific; the fasting glucose test is specific but not sensitive'.[15] Holman et al (p.6) pointed out, however, that 'Although HbA1c and fasting identify different groups of people with undiagnosed diabetes, the proportion of people that are identified is similar'.[18]

Our review aimed to answer two questions about the models.

1. How were the models validated? A basic test of the validity of a forecasting model is to apply this to past data to predict a known future for example, does the model using HSE data from 2004 predict prevalence as estimated from HSE data in 2014? None of the accounts of the models we reviewed reports such a test. The PHE model[5] was validated by refitting the model on 70% of the data (randomly selected) and checking its estimates against the remaining 30% of data.

2. Did the models try to take account of future changes in age-specific prevalence rates? Only the APHO model[18] aimed to do this by estimating the net effect of trends in: changes in ethnicity; and being overweight and obese to create a sex-specific obesity adjustment index. They did not, however, give details of how that index was modelled.[5 21 26] The other three models[5 21 26] assumed that future age-specific prevalence of diabetes would be as estimated from past HSEs.

The prevalence-based models we reviewed are focused on estimating geographical variations in the future prevalence of diabetes within countries, rather than giving sound estimates of future totals.

### Rapid review 2: Markov chain models

Rapid review 2 of Markov models identified 304 articles. An additional one was snowballed. After removing duplicates, 222 articles were screened, 20 of them were considered relevant and fully assessed. Of these, one was excluded because we could not locate it, one did not report the results, and one modelled the progression from diabetes to its complications only. Table 2 gives details of the remaining 17 articles,[27–43] ordered in terms of their completeness of the information we could find on transition probabilities. (online supplementary appendix 2 gives additional information on objectives, model, population, outcomes, sensitivity analysis and validation.) Two articles did not report the measure of IH used.[39 43] Twelve reported a model using one risk measure only: nine models used IGT,[28 29 31–34 37 38 42] two HbA1c[36 40] and one FPG only.[27] Neumann et al reported two models, using IFG and IGT[28]; and Roberts et al,[35] three models using HbA1c, IGT and IFG. Hence, we reviewed 20 models.

**Table 2** Transition probabilities reported in different models for no preventive intervention (or standard care)

| Reference | Measure of Intermediate Hyperglycaemia (IH) | Country | Normoglycaemia (NG) to IH | IH to NG | NG to T2D | T2D to NG | IH to T2D | T2D to IH | Mortality rates (relative risk*) |
|---|---|---|---|---|---|---|---|---|---|
| Johansson et al,†[27] | FPG | Sweden | | | | | | | |
| Herman et al[38] | IGT | USA | | | | | 10.80%[49] | | |
| Palmer et al[37] | IGT | Australia, France, Germany, Switzerland and UK | | | | | Overall 11% for standard care varies by age (10.8% to 11.6%) and body mass index (9.0% to 14.3%)[50] | | IH:1.37 (1.05 to 1.79) Undiagnosed T2D: 1.76 (1.17 to 2.66) Diagnosed T2D: 2.26 (1.78 to 2.87) |
| Zhuo et al†[40] | HbA1c | USA | | | | | 0.07% to 18.9% by HbA1c[51] | | |
| Chen et al[39] | | Taiwan | | | 1.10%[52] | | | | |
| Zhou et al[36] | HbA1c | USA | | | 0% | 0% | | 0% | |
| Schaufler and Wolff[41] | IGT or IFG | Germany | male, 2.23% female, 1.45%[53] | | Male, 2.51% and female, 1.66%[53] | | Male, 4.79% female, 4.23%[53] | | Source given for higher mortality rates for T2D[54] |
| Gillies et al[29] | IGT | UK | <65, 1.66% >65, 2.49%[55] | | | | 1.96% based on 12 studies[55–66] | | Increased risk of death with diabetes (HR) 0.756 (SE=0.087)[67] 1% increase in HbA1c (HR) 0.104 (SE=0.039)[68] |
| Palmer and Tucker[30] | IGT | Australia | | Reported over time for standard care[69] ▲ 10%, year 1 ▲ 5.6% year 2 ▲ 3.5% years >2 | Reported for standard care 4.6%[55] | 0% | Reported over time for standard care ▲ 11%, years 1 to 3[50] ▲ 5.6%, years >3[50] | 0% | IH: 1.50 (1.10 to 2.00) 'undiagnosed' T2D: 1.30 (0.90 to 2.66) 'diagnosed' T2D 2.30 (1.60 to 3.20)[70] |
| Ikeda et al[42] | IGT | Japan | 3.10%[71] | For standard care 33.1%[72] | 0% | 0% | For standard care 6.6%[72] | 0% | IH: 1.35 T2DM: 3.03[73] |
| Smith et al[43] | | USA | 4%[74] | | 0.40%[75] | 0% | 10.80%[38] | 0% | IH: 1.7[76] stable T2D: 2[77] complicated T2D: 2.4[78] |

Continued

**Table 2** Continued

| Reference | Measure of Intermediate Hyperglycaemia (IH) | Country | Normoglycaemia (NG) to IH | IH to NG | NG to T2D | T2D to NG | IH to T2D | T2D to IH | Mortality rates (relative risk*) |
|---|---|---|---|---|---|---|---|---|---|
| Neumann et al[28] | IGT | Sweden | Risk equation reported | Risk equation reported | 0% | 0% | Risk equation reported | Risk equation reported | No increased risk for IH. T2D mortality not reported. |
| Caro et al[31] | IGT | Canada | 16.30% (original estimate) | 16.20% (original estimate) | 0% | 0% | 6.30% (original estimate) | 0% | IH: 1.45 (original estimate) |
| Neumann et al[32] | IGT | Germany | 16.30%[31] | 16.20%[31] | 0% | 0% | 6.00%[79] | 0.50% (original estimate) | |
| Liu et al[33] | IGT | China | 1.28%[80] | 11.60%[81] | 0% | 0% | Initiation ages 25: 6.44% 40: 16.7% 60: 57.8%[82–84] | 0% | |
| Wong et al[34] | IGT | Hong Kong | 16.30%[31] | 16.20%[31] | 0% | 0% | For usual practice, years 1 to 3, 11%;[58] years >4, 5.6%[50] | 0% | IH: 1.50 (1.10 to 2.00) T2D: 2.30 (1.60 to 3.20)[30] |
| Roberts et al[35] | IGT | England | 6.33%[55] | 8.97%[85] | 0% | 0% | 4.55%[44] | 0% | IH: 1.50 T2D: 1.9[86] |
| | HbA1c | England | 6.86%[55] | 8.97%[85] | 0% | 0% | 3.55%[44] | 0% | IH: 1.2 T2D: 1.6[86] |
| | IFG (ADA) | England | 6.86%[55] | 8.97%[85] | 0% | 0% | 4.74%[44] | 0% | IH: 1.2 T2D: 1.6[86] |
| Range (for single probabilities) | IGT | | 1.28%–16.30% | 8.97%–16.20% (and for standard care from 3.5% to 33.1%) | 0.00%–2.5% (male) (and 4.6% for standard care) | 0% | 1.96%–10.8% (and 11% for standard care) | 0.00%–0.5% | IH:1.35 to 1.7 T2D: 1.76 to 3.03 |
| Meta-analyses | IGT | | | | | | 4.55%[44] | | IH: 1.32 (1.23 to 1.40)[11] |
| | HbA1c | | | | | | 3.55%[44] | | IH: 0.97 (0.88 to 1.07)[11] |
| | IFG (ADA) | | | | | | 3.54%[44] | | IH: 1.13, (1.02 to 1.25)[11] |

0%: not allowed.
*Relative risk over NG specified in,[28 29 31 34 42] ranges in parentheses are 95% CIs.
†Models described elsewhere.
FPG, fasting plasma glucose; HbA1c, glycated haemoglobin; IGT, impaired glucose tolerance; T2D, type 2 diabetes.

Our objective was to develop a matrix of transition probabilities, with one transition probability only between states, and hence designed to use available data for England. Table 2 gives the transition probabilities we found and shows no article provided the complete matrix of transition probabilities. Five only reported the full set between states other than death. No article reports transition probabilities from different states to death (ie, mortality rates for each state) and, where relative risk of mortality is reported for IH and T2D, we could not always find whether this was compared with normoglycaemia. Nor could we find how these models satisfied the fundamental requirement of a Markov chain model that all transition probabilities out of a state, estimated from different datasets, (including return to that state) sum to one.

Our review aimed to answer three questions about the Markov chain models:

1. Do these articles provide evidence of the likely impact of national preventive programmes? The primary focus of the articles we reviewed is on estimating the ratio of costs to benefits of preventive interventions for those who are hyperglycaemic (most based on IGT, only three for HbA1c, two for IFG and one for FPG). None reported the impact of preventive interventions on reducing the burden of disease from T2D in the general population. Only four articles[19 25 31 40] modelled the general population (with normoglycaemia and IH).

2. How were the models validated? Whereas most articles reported outcomes of sensitivity analyses, only five reported comparisons of their models' outputs with other empirical data: clinical trials[23 32]; the population with T2D in southern Wisconsin[36]; the disease progression of T2D in Germany[33]; mortality data for England and estimates of current prevalence of T2D by age group.[27] A good empirical test of a model's validity is of its estimates of those developing T2D in the absence of a preventive intervention. Only Caro et al[31] reported this for a general population, but they did not report a check against other projections. Of the articles that modelled populations with IH, only three reported estimates of the percentages developing T2D in the absence of intervention.[15 23 24]

3. How do transition probabilities compare? All models, except that of Neumann et al,[32] allow transitions from T2D to T2D or death only. Neumann et al[32] allow transition (at a low probability, 0.5%) from T2D to IH (IGT) (because 'this transition exists but seldom occurs', p

4). Only two models allow transition from normoglycaemia directly to T2D: Schaufler and Wolff[41] (IFG or IGT—for males, 2.51% and females, 1.66%) and Smith et al (measure of IH not specified, 0.40%).[43] For the transition probabilities reported in table 2, two models allow for changes over time[23 26]; and seven for variations by age.[20–23 25 29 30] Table 2 shows that wide ranges of transition probabilities used by the different IGT models: from normoglycaemia to IH, 1.28%–16.30%; from IH to low, 8.97%–16.20%; normoglycaemia to T2D, 0.00%–2.51% (for males); IH to T2D, 1.96%–10.8%. A meta-analysis recommended a rate of 4.55% for the last.[44]

The relative risks reported for IH for IGT ranged from 1.35 to 1.7; and T2D from 1.76 to 3.03. Roberts et al[35] report these risks for HbA1c to be 1.2 and 1.6. The estimates from the systematic review and meta-analysis[11] for IH were: for IGT 1.32 (1.23–1.40) and for HbA1c 0.97 (0.88–1.07). One article[32] reported a matrix in which probabilities of transitions between states other than death sum to one, which implies no one dies. PHE defines those with IH using either HbA1c or IFG.[5] The models developed by Roberts et al[35] for HbA1c and IFG are similar. We used their HbA1c model to project the prevalence of T2D in England. They used the recommended transition probabilities from different risk measures of IH to T2D identified by a meta-analysis.[44] Neumann et al[32] and Caro et al[31] have similar transition probabilities, which are higher than those of Roberts et al,[35] for IGT from normoglycaemia to IH, and IH to T2D: 16.3% and 6.00% compared with 6.33% and 4.55%. We used the transition probabilities used by Neumann et al[32] because that is more recent. Model 1 is based on Roberts et al (HbA1c),[35] which was modified as model 2 to give the projections of PHE. To do this, model 2's transition probability from IH to T2D (0.013) is a third of that of model 1 (0.036), and below the lowest rate of any model we reviewed (0.02). (Model 2 has a corresponding increase in the transition probability of remaining in IH (0.836 to 0.878)). Model 3 is based on Neumann et al.[32] Details of the models are given in online supplementary appendix 1.

## Estimating the trend in diagnosed diabetes

Table 3 reports the OLS estimate of the trend in diagnosed diabetes from QOF data,[17] which gives an annual rate of increase of 0.11%.

**Table 3** The trend model from QOF data

| Coefficients | Value | SE | T | Pr > |T| | Lower bound (95%) | Upper bound (95%) |
|---|---|---|---|---|---|---|
| Intercept | −219 | 4.375 | -50.14 | <0.0001 | −210 | −229 |
| Year | 0.110 | 0.002 | 50.71 | <0.0001 | 0.105 | 0.115 |
| Adjusted R squared | 0.995 | | | | | |

QOF, Quality and Outcomes Framework.

### Comparing projections of the future prevalence of T2D

Table 4 gives: for the different prevalence-based models their defined populations, data sources and projections of diabetes true prevalence (in millions); comparable estimates of the true prevalence of diabetes from the QOF data and estimated trend (increased by a third) and the annual rate of increase in prevalence from the first in the series to the last. Table 4 shows that for the three models that do not allow for increase in prevalence rates by age and sex,[21–23] the older the HSE data used, the lower is the estimate of the rate of increase in prevalence for England. We compare projections of true prevalence of diabetes and T2D by different models giving numbers in millions; and, in parentheses, CIs (where available).

Global models give three projections of the true prevalence for diabetes prevalence in the UK (aged 20–79): for 2030, 2.55[21] and 3.65[22] and 2035, 3.62.[23] Each projection is below the estimate by PHE[12] for England for 2015, 3.81 (based on HSEs for 2012, 2103 and 2014). These global models assume low rates of increase in prevalence over time and exclude those over 79, who we estimated to account for over 25% of the number who would be aged 20 to 79 in England and develop diabetes in 2030 and 2035. The projections by these global models are not examined further.

Two models give projections of the true prevalence of diabetes for England only (aged over 15): the PHE model[12] for 2030, 4.68 and 2035, 4.94 and APHO for 2030, 4.60 (3.25 to 6.88).[18] The two accounts of the APHO model[15 16] report the same projection for 2030; but one estimated the prevalence of diabetes in 2010 (3.10)[18] to be higher than the other for 2013 (2.17).[16] And the increase in prevalence to 2030 attributed to increases in obesity was estimated to be a half[18] and a third.[25] Figure 4 compares three projections for 2025: PHE,[12] 4.39; Holman et al 4.19 (2.93–6.19); and the QOF trend, 5.46 (5.32–5.59), which has a narrow CI because this trend has been so stable.

Figure 5 compares projections of the true prevalence of T2D in England to 2035, the QOF trend and our three Markov chain models. This shows that the projections by model 2 replicated the projections by PHE; by model 1 are above those from PHE and the QOF trend; by model 3 seem to be implausibly explosive. Figure 5 also shows the impact of reducing the estimate of those with IH to zero in 2015 on the projections by models 1 and 3. Table 5 gives projections for 2025, These are: 3.95, by PHE; 4.91 (4.79–5.03) from the QOF trend; 5.64 by model 1; 3.86 by model 2 and 9.07 by model 3. Putting the estimate of those with IH to zero in 2015 reduces the projections by models 1 and 3 to 5.01 and 8.57, which are above the projections by PHE and the QOF trend.

### DISCUSSION

Akushevich et al[45] point out that although the 'prevalence probability of a disease is a fundamental epidemiological characteristic' for which there are various data sources, this random variable is the difference between changes

**Table 4** True diabetes prevalence (millions) estimated by different epidemiological models and from the QOF trend

| Source of estimate | Details of series | | | | | | Mean annual increase (%)* | Projections | | | | |
|---|---|---|---|---|---|---|---|---|---|---|---|---|
| | Population | Data source | First year | Prevalence | Final year | Prevalence | | 2015 | 2020 | 2025 | 2030 | 2035 |
| Shaw et al[21] | UK: 20–79 (UN, 2007) | HSE (2003) | 2010 | 2.14 | 2030 | 2.55 | 0.02 | | | | 2.55 | |
| Whiting et al[22] | UK: 20–79 (UN, 2011) | HSE (2004 and 2009) | 2011 | 3.06 | 2030 | 3.65 | 0.031 | | | | 3.65 | |
| Guariguata et al[23] | UK: 20–79 (UN, 2011) | HSE (2004) | 2013 | 2.98 | 2035 | 3.62 | 0.029 | | | | | 3.62 |
| Holman et al[18] | England: >15 (ONS) | HSE (2006) | 2010 | 3.10 | 2030 | 4.60 (3.25–6.88) | 0.075 | 3.47 (2.47–5.07) | 3.82 (2.70–5.62) | 4.19 (2.93–6.19) | 4.60 (3.25–6.88) | |
| PHE[5] | England: >15 (ONS) | HSE (2012, 2013 and, 2014) | 2015 | 3.81 | 2035 | 4.94 | 0.056 | 3.81 | 4.09 | 4.39 | 4.68 | 4.94 |
| QOF data and trend† | England: >15 registered with GPs | QOF (2004–2005 to 2017–2018) | 2004–2005 | 2.36 | 2017-18 | 4.26 | 0.147 | 3.99 (3.88–4.09) | 4.72 (4.61–4.84) | 5.46 (5.32–5.59) | 6.19 (6.04–6.35) | 6.93 (6.75–7.11) |

*Estimated as the annual mean increase from the first estimate to the last.
†To estimate the true prevalence from the QOF data and trend both sets of estimates were increased by a third. They are based on data from 2004-5 to 2017-18 and the estimated trend from 2015 to 2035.
GP, general practitioner; HSE, Health Surveys for England; PHE, Public Health England; QOF, Quality and Outcomes Framework.

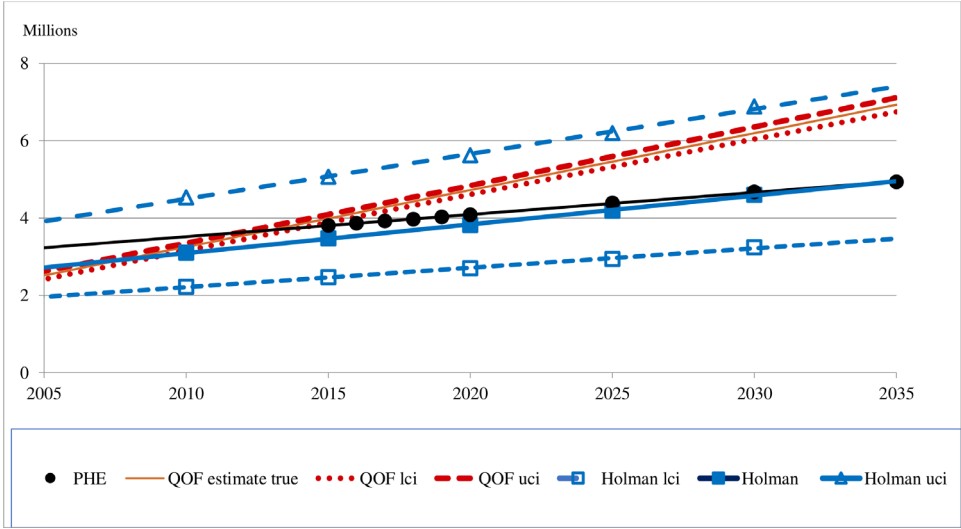

**Figure 4** Projections of true diabetes prevalence in England: 2005–2035. PHE, Public Health England; QOF, Quality and Outcomes Framework.

over time in disease incidence and patient survival. This has a statistical implication that, whatever modelling approach is used, we would expect projections of prevalence to have large errors of estimation. The policy implication, which Akushevich *et al* emphasise, is that the overriding objective ought to be to improve population health, rather than reducing the prevalence of T2D: because, for example, improving survival of those with T2D may increase prevalence (depending on changes in incidence). Akushevich *et al* developed a new methodological approach that partitions trends in observed disease prevalence into their two components, and hence gives estimates of the direction and strength of the effect of each. Their models are estimated from a single data set (Medicare data), incorporate changes over time and take account of age.

The four prevalence-based models we reviewed[10 14 15 17] use past estimated prevalence rates by age and sex and projected changes in populations. They are focused on estimating geographical variations in the future prevalence of diabetes within countries, rather than giving sound estimates of future totals. Only one model aims to take account of changes in prevalence rates by age and sex over time.[15] Of the five projections of diabetes prevalence, for England and the UK we reviewed,[10–12 14 15] only one[15] reported CIs.

The Markov chain models of the economic impacts of interventions that aim to prevent T2D, which we reviewed, aim to capture changes in incidence and survival in one model. Their primary focus is on estimating the ratio of costs to benefits of preventive interventions for those who are hyperglycaemic (mostly based on IGT). None reported the impact of preventive interventions on reducing the burden of disease from T2D in the general population. We could not find a complete matrix of transition probabilities; nor descriptions of how transition probabilities estimated from different datasets satisfied the fundamental requirement of a Markov chain

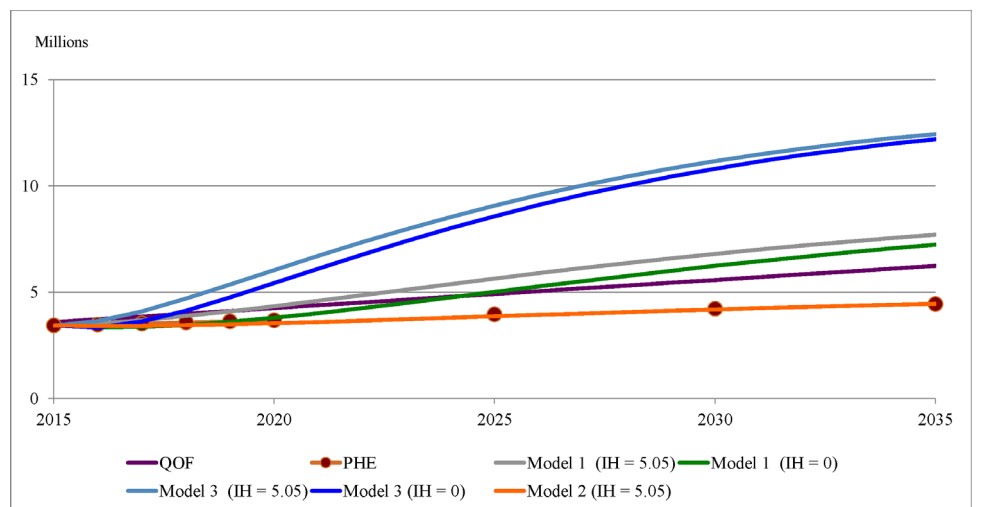

**Figure 5** Projections of the true prevalence of T2D in England: 2015–2035. PHE, Public Health England; QOF, Quality and Outcomes Framework; T2D, type 2 diabetes.

**Table 5** Projections of the true prevalence of T2D in England for 2025

| Model | Projections for 2025 (millions) | | | |
|---|---|---|---|---|
| | Statistical | | Markov (numbers with intermediate hyperglycaemia in 2015) | |
| | Point estimate | 95% CIs | 5.05* | Zero |
| PHE | 3.95 | n.a. | | |
| QOF trend | 4.91 | 4.79 to 5.03 | | |
| Model 1† | | | 5.64 | 5.01 |
| Model 2‡ | | | 3.86 | |
| Model 3§ | | | 9.07 | 8.57 |

*As estimated by PHE.
†Based on Roberts et al.[35]
‡Based on Roberts et al,[35] but modified to reproduce the PHE trend to 2035.
§Based on Neuman et al.[32]
n.a., not available; PHE, Public Health England; QOF, Quality and Outcomes Framework; T2D, type 2 diabetes.

model that all transition probabilities out of a state sum to one. The transition probabilities we did find do not vary over time. In seven articles these probabilities do vary by age.[20–23 25 29 30] In their systematic review of models of the economic impacts of preventive interventions, Leal et al[7] also found the majority of models assumed that 'the rate of progression to T2D was constant across the entire pre-diabetes population'. They attribute this in part to limitations in the available data, but highlight the 'stark contrast' between these simple models and 'The complexity of risk prediction models for diabetes incidence and the variety of covariates used'.[46 47] Friedman famously[48] argued, however, that the relevant question to ask about the 'assumptions' of economic theory, 'is not whether they are descriptively realistic … but whether the theory works, which means that it yields sufficiently accurate predictions' (p 153).

Three projections of diabetes prevalence (in millions) for the UK (aged 20–79) by global models are: for 2030, 2.55[21 22] and 3.65,[21 22] and for 2035, 3.62.[23] Each is below the PHE estimate of 3.81 for 2015 for England only (over 15).[12] This raises questions over the validity of these global projections; and their excision of those over 79, who we estimated to account for over 25% of the number who would be aged 20 to 79 and develop T2D after 2030. We report three estimates of diabetes prevalence in England for 2025 (with 95% CIs where available): 4.39 by PHE,[12] 4.19 (2.93 to 6.19) by the APHO model,[18 25] and 5.46 (5.32 to 5.59) from the QOF trend. We and Leal et al[7] found only minority of articles reported tests of validation. Such checks are vital for Markov chain models given the different data sources used to estimate transition probabilities.

Our Markov chain models are based on transition probabilities to states other than death from published models, to death from English mortality rates, and of remaining in a state as the residual (so all transition probabilities from each state sum to one). The projections of prevalence of T2D for England for 2025 are: 5.64 by model 1 (based on Roberts et al for HbA1c),[35] and 9.07 by model 3 (based on Neuman et al for IGT).[32] To reproduce PHE's projections by model 2, of 3.86, model 1 was modified with a lower probability of transition from IH to T2D than any of the models we reviewed. These comparisons suggest that the PHE projection of T2D prevalence in 2025 of 4 million is too low, and a more realistic estimate is about 5 million.

The limitations of our research are our models are simple and transparent, and, as we did not undertake systematic reviews, we may have omitted relevant articles. The systematic review by Leal et al[7] reviewed 29 studies, which included 12 of the 17 studies of Markov chain models that we reviewed. Their principal findings are strikingly similar to ours. They recommend the development of 'more comprehensive models that are capable of better capturing the continuity in disease progression and, also, of incorporating the identification of novel biomarkers'. But, they recognise such models require more detailed data and only need to be comprehensive enough to provide reliable estimates for decision making.

## CONCLUSIONS

There are three implications of our reviews of two types of models used to project prevalence of T2D. First, current prevalence-based models are focused on estimating geographical variations in the future prevalence of diabetes within countries, rather than giving sound estimates of future totals. They are designed to underestimate the scale of increases in the future prevalence of T2D in England and the UK, and hence the urgency for governments to implement preventive interventions. Second, the primary focus of the Markov chain models is on estimating the ratio of costs to benefits of preventive interventions for those who are hyperglycaemic (mostly based on IGT). We found that no articles gave the complete matrix of transition probabilities and a full description of how they were derived. Only a minority have been subjected to tests of validity. Third, to inform national policies, governments need estimates of the impacts of preventive interventions on reducing the burden of disease from T2D in the general population. These estimates ought to be derived from validated models, designed to use available data, that estimate changes over time in the incidence and survival of patients with T2D, with and without preventive interventions.

**Acknowledgements** We are grateful to our two referees, Anders Green and Igor Akushevich, for critical comments on earlier drafts. Their comments have clarified our argument and helped us to consider the adequacy of two types of models that project changes in prevalence, given that this is the observed outcome of changes in incidence and survival.

**Contributors** MJK did the original work in developing initial Markov Chain models to estimate the impacts of preventive interventions on the future prevalence of Type 2 Diabetes (T2D) in England and has been involved throughout this project. CDP worked with MJK in developing those models, undertook the rapid reviews of epidemiological and Markov Chain models, and commented on drafts of this paper. GB prepared drafts of the paper, reviewed epidemiological and Markov Chain models, developed the models used in this paper and undertook comparisons of projections. RR took part in the workshops on the models, reviewed our methods and findings, and commented on drafts.

**Funding** This study was funded by the National Institute for Health Research (NIHR) Collaboration for Leadership in Applied Health Research and Care North Thames at Barts Health NHS Trust, which had no role in the writing of the manuscript or the decision to submit it for publication.

**Competing interests** None declared.

**Patient consent for publication** Not required.

**Provenance and peer review** Not commissioned; externally peer reviewed.

**Data availability statement** All data relevant to the study are included in the article or uploaded as online supplementary information. The data we have used are from cited public sources.

**ORCID iDs**
Gwyn Bevan http://orcid.org/0000-0003-2123-3770
Chiara De Poli http://orcid.org/0000-0002-1879-553X
Mi Jun Keng http://orcid.org/0000-0001-5979-1706
Rosalind Raine http://orcid.org/0000-0003-0904-749X

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
