## [Reviewer comments · BMJ Open]

ARTICLE DETAILS

TITLE (PROVISIONAL)	How valid are projections of the future prevalence of diabetes? Rapid reviews of prevalence-based and Markov chain models and comparisons of different models' projections for England
AUTHORS	Bevan, Gwyn; De Poli, Chiara; Keng, Mi; Raine, Rosalind

VERSION 1 - REVIEW

REVIEWER	Anders Green Open Patient Data Exploratory Network (OPEN) Odense University Hospital and University of Southern Denmark, Denmark
REVIEW RETURNED	14-Sep-2019

GENERAL COMMENTS	The paper reviews aspects of projecting the future prevalence of diabetes from existing data. This is of high importance not only for diabetology but also because the methodologies may be applied to all chronic diseases as well. The paper is difficult to read and gives rise to some comments. Major comments: 1. The present reviewer strongly disagrees to use the value-laden label 'epidemiological models' exclusively for methods relying on projections of prevalence estimates and demography. This is not fair to epidemiology, and even epidemiologists have applied Markov chain models to describe observed and predict future trends in the prevalence of diabetes. It is suggested to label the two classes of models considered as 'Prevalence based models' and 'Markov chain models', respectively.2. It is relevant and justified to emphasize the public health importance and the prevention aspects in particular when predicting the future prevalence of diabetes. In this respect it is a major shortcoming that the paper ignores important drivers of prevalence. Certainly, the prevalence is driven by incidence and mortality. But the mortality in diabetes depends heavily on complication state, and the risk of complications as well as the mortality given complication state will change over time, thereby impacting trends in the mortality. This is the main reason why prediction models based on simple projections of prevalence in general yield under-estimates of the future prevalence. There may not be sufficient UK data to populate a Markov chain that also include states of complications (like kidney disease, heart disease, neurological and vascular complications presenting as amputations) between diagnosis and death, but at least this aspect should be discussed appropriately in the paper.
---

	3. It is recommended to scrutinize the paper to improve wording and clarity. Some examples: a. P. 7, lines 38-43: It is not intuitively clear for the reader what the hazard ratios mentioned refer to. What is the reference category for these ratios? b. P. 8, line 37 & p. 10, line 44: What does 'snowballed' mean in the present context? c. P. 9, line 52: 'Fasting glucose test' is mentioned twice in the sentence, with different characteristics attached. Is this a misquotation of the paper referenced? d. Throughout the paper (exemplified on p. 15 and in Table 2) it seems that transition rates and transition probabilities are used interchangeably. This is inconsistent from an epidemiological point of view: Rates denotes number of events observed per time unit with the dimension per time-out. In contrast, probabilities are dimensionless. For rare events, the estimated rate will have a value similar to the probability of developing the event over the time-unit used in the rate. But for more frequent events the value of the probability will be less than the value of the underlying rate. It is recommended either to use only one of these quantities throughout the papers or making it clear for the reader when probability is used and when rate is used.
--	--

REVIEWER	Igor Akushevich Duke University, United States
REVIEW RETURNED	21-Sep-2019

GENERAL COMMENTS	The objective of the manuscript entitled "How valid are models' projections of the future prevalence of diabetes? Rapid reviews of epidemiological and Markov chain models and comparisons of different models' projections for England" is formulated as "to examine validity of epidemiological models giving projections of prevalence of diabetes in adults, in England and the UK, and of Markov chain models giving estimates of impacts of interventions to prevent type 2 diabetes". The authors reviewed available approaches for modeling and projections of diabetes prevalence which involve epidemiological and Markov chain models. The authors also suggested their own approach based on a Markov chain modeling approach. Unfortunately I did not find a clear answer on the question formulated in the title. I would expect that I will be able to find the answer in Discussion section, but the authors did not formulate their own conclusion about which model should be further used to better predict diabetes prevalence. To clearly understand which model projections is preferable, a reader have to see, at least, a statement of the author about that, clear explanation why the suggested model is better than others, brief review of the technical approach underlying the projection construction, and discussion of limitations with explanations of how to address them. The author focused their discussions on the properties of transition probabilities. However, the main critical properties of such probabilities, (e.g., their dependence on patient age and other individual characteristics) are not sufficiently clarified and discussed. For example, in the model the authors suggest all transition probabilities are just constants. Does it mean that
--

	respective rates of diabetes incidence are age-independent? Are these really realistic assumptions? The transition probabilities represent the population that was used for their estimation. I would expect more discussion of these and respective model limitations, at least for the best model(s). The authors restricted their analyses to a class of the models that are transparent and simple. Specifically they focused on epidemiological and Markov chain models. Readers should understand whether the chosen class of model is main, and other models are not appropriate because of some reasons, or alternatively, other models that sometimes suggest a quite rigorous modeling approach (e.g., Mathematical Biosciences 311 (2019) 31–38) could be also helpful. Discussion of the place of the chosen and considered class of models among all available models dealing with predictions of diabetes prevalence and outcomes would be helpful for readers.
--	--

VERSION 1 – AUTHOR RESPONSE

Professor Anders Green

Comment

1. The present reviewer strongly disagrees to use the value-laden label 'epidemiological models' exclusively for methods relying on projections of prevalence estimates and demography. This is not fair to epidemiology, and even epidemiologists have applied Markov chain models to describe observed and predict future trends in the prevalence of diabetes. It is suggested to label the two classes of models considered as 'Prevalence based models' and 'Markov chain models', respectively.

Response

We designed our search of the literature to identify those models giving estimates of the future prevalence of diabetes. We did not restrict our search to prevalence-based models. Epidemiologists may indeed use Markov chain models, as we have done, to project future prevalence, but we found no articles that did so -- we found only prevalence-based models. Our search terms included "epidemiolog*" OR "prevalence" OR "incidence" OR "trend*" (given in Table 1.1 of Appendix 1). Furthermore, we would expect 'epidemiological models' to be based on prevalence rates, the weakness of three of the four models we reviewed was that they did not take account of changes in prevalence rates by age and sex in making future projections.

Comment

2. Mortality of those with diabetes depends heavily on complications and risk of complications. The mortality rate given complications will change over time. This omission is the main reason why prediction models based on simple projections of prevalence in general yield under-estimates of the future prevalence. There may not be sufficient UK data to populate a Markov chain that also include states of complications (like kidney disease, heart disease, neurological and vascular complications presenting as amputations) between diagnosis and death, but at least this aspect should be discussed appropriately in the paper.

Response

This comment by Professor Green relates to those by Professor Akushevich. Both argue that more complex models are required. We give our response to this comment by Professor Green at the end of our responses to comments by Professor Akushevich.

Comment

3. Improve wording and clarity:

Comment

- a. P. 7, lines 38-43: It is not intuitively clear for the reader what the hazard ratios mentioned refer to. What is the reference category for these ratios?

Response

We have explained that these ratios are with reference to those with normoglycaemia.

Comment

- b. P. 8, line 37 & p. 10, line 44: What does 'snowballed' mean in the present context?

Response

What 'snowballed' means is as described by Olin et al (2014): 'Snowballing refers to using the reference list of a paper or the citations to the paper to identify additional papers'.

We have revised our sentence as follows: "Rapid review 1 of methods of epidemiological models retrieved 633 articles and from their citations we identified a further five by snowballing (Olin et al, 2014)."

Comment

- c. P. 9, line 52: 'Fasting glucose test' is mentioned twice in the sentence, with different characteristics attached. Is this a misquotation of the paper referenced?

Response

We have corrected this error. The quotation now reads: 'The most commonly used test (HbA1c) is neither sensitive nor specific; the fasting glucose test is specific but not sensitive'.

Comment

- d. The paper ought to use transition probabilities throughout and not transition rates.

Response

We have made this change.

Professor Igor Akushevich

Comment

1. I did not find a clear answer on the question formulated in the title. The authors did not formulate their own conclusion about which model should be further used to better predict diabetes prevalence. To clearly understand which model projections is preferable, a reader have to see, at least, a statement of the author about that, clear explanation why the suggested model is better than others, brief review of the technical approach underlying the projection construction, and discussion of limitations with explanations of how to address them.

Response

The original title of our paper was: "How valid are models' projections of the future prevalence of diabetes? Rapid reviews of epidemiological and Markov chain models and comparisons of different models' projections for England". Our paper considers projections by different models, their validity and makes comparisons. It does not aim to recommend any of these models given the concerns we have about their validity.

Comments

2. The author focused their discussions on the properties of transition probabilities. However, the main critical properties of such probabilities, (e.g., their dependence on patient age and other individual characteristics) are not sufficiently clarified and discussed. For example, in the model the authors suggest all transition probabilities are just constants. Does it mean that respective rates of diabetes incidence are age-independent? Are these really realistic assumptions? The transition probabilities represent the population that was used for their

estimation. I would expect more discussion of these and respective model limitations, at least for the best model(s).

3. The authors restricted their analyses to a class of the models that are transparent and simple. Specifically they focused on epidemiological and Markov chain models. Readers should understand whether the chosen class of model is main, and other models are not appropriate because of some reasons, or alternatively, other models that sometimes suggest a quite rigorous modeling approach (e.g., *Mathematical Biosciences* 311 (2019) 31–38) could be also helpful. Discussion of the place of the chosen and considered class of models among all available models dealing with predictions of diabetes prevalence and outcomes would be helpful for readers.

Response to comments by Professor Green (2) and Professor Akushevich (2 and 3)

Professors Green and Akushevich are correct that the two types of models we reviewed are transparent and simple. Each referee recognises the problem of lack of routinely-available data for more complex models. Airoidi et al (2008), for example, report their development of a model of the impacts of better glucose control in adolescents for Type 1 Diabetes (T1D) in England that took account of complications of diabetic nephropathy and retinopathy and diabetic foot. Their objective was to develop a requisite decision model, which is defined as 'A model whose form and content are sufficient to solve a particular problem' (Phillips, 1984). Airoidi et al aimed to develop a model that would give reliable estimates (in terms of orders of magnitude) for informing strategic commissioning by being transparent and simple and using routinely-available data. We cite this study to make two points. First, the lack of data required Airoidi et al to make assumptions on: incidence of T1D; prevalence of T1D by age group; prevalence of degrees of severity of renal and eye disease complications; transition probabilities for those with diabetic nephropathy and retinopathy (excluding mortality rates); mortality rates for those with diabetic nephropathy and retinopathy; incidence rates of amputations, sores or ulcers; and mortality rates in the non-diabetic population. Second, they tested the adequacy of their model through tests of validation (of estimates of diabetic nephropathy and retinopathy and diabetic foot), sensitivity analysis and comparing results with those from more sophisticated models. The models we reviewed have desirable attributes for informing policy on preventing T2D by being simple and transparent and designed to use routinely-available data. Our concern is whether they are requisite: i.e. do they give reliable estimates in terms of orders of magnitude? Both classes of model we reviewed often lack of tests of validity and the differences in projections of the future prevalence of T2D differ by orders of magnitude.

So, we have added this paragraph to open our concluding section:

The epidemiological and Markov chain models we reviewed have desirable attributes for informing policy on preventing T2D by being simple and transparent and designed to use routinely-available data. The Markov Chain models, for example, do not take account of diabetic complications or age. We have considered whether both types of models requisite in their form and content (Phillips, 1984) for the objective of giving reliable estimates of the order of magnitude of the future prevalence of T2D. We conclude that they are not. This is because both types of models often lack of any tests of validity, and the differences in projections of the future prevalence of T2D differ by orders of magnitude.

Professor Akushevich asks about the importance of the papers we have reviewed. We have changed the introduction to our methods section as follows:

Our comparisons of projections of different models builds on two reviews of the literature, which were designed to be rapid (not systematic) by using stringent criteria to identify the principal methods of each type of models.

We have examined the number of citations of these papers using Google Scholar (on 1 October 2019) and the results are given in Table 1. This shows there were over 14,000 citations of the epidemiological models, and over 800 citations of the Markov chain models. Hence each represents important fields of research into estimating the future prevalence of T2D and the cost-effectiveness of preventive interventions.

Table 1: Numbers of citations of papers on epidemiological models reviewed in Google Scholar (1 October 2019)

Reference	Number of citations
Epidemiological models	
Shaw JEE, Sicree RAA, Zimmet PZZ. Global estimates of the prevalence of diabetes for 2010 and 2030. Diabetes Res Clin Pract 2010;87:4–14.	7,169
Whiting DR, Guariguata L, Weil C, et al. IDF diabetes atlas: global estimates of the prevalence of diabetes for 2011 and 2030. Diabetes Res Clin Pract 2011;94:311–21	3,911
Guariguata L, Whiting DR, Hambleton I, et al. Global estimates of diabetes prevalence for 2013 and projections for 2035. Diabetes Res Clin Pract 2014;103:137–49	3,239
Holman N, Forouhi NG, Goyder E, et al. The Association of Public Health Observatories (APHO) Diabetes Prevalence Model: estimates of total diabetes prevalence for England, 2010-2030. Diabet Med 2011;28:575–82.	109
Guariguata L, Whiting D, Weil C, et al. The International Diabetes Federation diabetes atlas methodology for estimating global and national prevalence of diabetes in adults. Diabetes Res Clin Pract 2011;94:322–32.	230
Public Health England. Technical document for the diabetes prevalence model for England 2016. London: 2016.	0
Total	14,658
Markov chain models	
Johansson P, Östenson C-G, Hilding AM, et al. A cost-effectiveness analysis of a community-based diabetes prevention program in Sweden. Int J Technol Assess Health Care 2009;25:350–8	25
Herman WH, Hoerger TJ, Brandle M, et al. The cost-effectiveness of lifestyle modification or metformin in preventing type 2 diabetes in adults with impaired glucose tolerance. Ann Intern Med 2005;142:323–32.	12
Palmer AJ, Roze S, Valentine WJ, et al. Intensive lifestyle changes or metformin in patients with impaired glucose tolerance: Modeling the long-term health economic implications of the diabetes prevention program in Australia, France, Germany, Switzerland, and the United Kingdom. Clin Ther 2004;26:304–21.	46
Zhuo X, Zhang P, Selvin E, et al. Alternative HbA1c Cutoffs to Identify High-Risk Adults for Diabetes Prevention. Am J Prev Med 2012;42:374–81.	28
Zhuo X, Zhang P, Selvin E, et al. Alternative HbA1c Cutoffs to Identify High-Risk Adults for Diabetes Prevention. Am J Prev Med 2012;42:374–81.	52
Zhou H, Isaman DJM, Messinger S, et al. A computer simulation model of diabetes progression, quality of life, and cost. Diabetes Care 2005;28:2856–63	113
Schaufler TM, Wolff M. Cost Effectiveness of Preventive Screening Programmes for Type 2 Diabetes Mellitus in Germany. Appl Health Econ Health Policy 2010;8:191–202.	37
Gillies CL, Lambert PC, Abrams KR, et al. Different strategies for screening and prevention of type 2 diabetes in adults: cost effectiveness analysis. BMJ 2008;336.	304

Reference	Number of citations
Ikeda S, Kobayashi M, Tajima N. Cost-effectiveness analysis of voglibose for prevention of type 2 diabetes mellitus in Japanese patients with impaired glucose tolerance. 2010;1:252–8	4
Smith KJ, Hsu HE, Roberts MS, et al. Cost-effectiveness analysis of efforts to reduce risk of type 2 diabetes and cardiovascular disease in southwestern Pennsylvania, 2005-2007. Prev Chronic Dis 2010;7:A109.	48
Neumann A, Lindholm L, Norberg M, et al. The cost-effectiveness of interventions targeting lifestyle change for the prevention of diabetes in a Swedish primary care and community based prevention program. Eur J Heal Econ 2017;18:905–19. doi:10.1007/s10198-016-0851-9	12
Caro JJ, Getsios D, Caro I, et al. Economic evaluation of therapeutic interventions to prevent Type 2 diabetes in Canada. Diabet Med 2004;21:1229–36.	79
Neumann A, Schwarz P, Lindholm L. Estimating the cost-effectiveness of lifestyle intervention programmes to prevent diabetes based on an example from Germany: Markov modelling. Cost Eff Resour Alloc 2011;9:17. doi:10.1186/1478-7547-9-17	35
Liu X, Li C, Gong H, et al. An economic evaluation for prevention of diabetes mellitus in a developing country: a modelling study. BMC Public Health 2013;13:729.	23
Wong CKH, Jiao F-F, Siu S-C, et al. Cost-Effectiveness of a Short Message Service Intervention to Prevent Type 2 Diabetes from Impaired Glucose Tolerance. J Diabetes Res 2016;2016:1–8.	7
Roberts S, Craig D, Adler A, et al. Economic evaluation of type 2 diabetes prevention programmes: Markov model of low- and high-intensity lifestyle programmes and metformin in participants with different categories of intermediate hyperglycaemia. BMC Med 2018;16:16.	17
Total	842

References

Airoldi, M., Bevan, G., Morton, A., Oliveira, M., & Smith, J. (2008). Requisite models for strategic commissioning: the example of type 1 diabetes. *Health care management science*, 11(2), 89-110.

Phillips, L. D. (1984). A theory of requisite decision models. *Acta psychologica*, 56(1-3), 29-48.

Wohlin, C., 2014, May. Guidelines for snowballing in systematic literature studies and a replication in software engineering. In *Proceedings of the 18th international conference on evaluation and assessment in software engineering* (p. 38). ACM.

VERSION 2 – REVIEW

REVIEWER	Anders Green Odense University Hospital Denmark
REVIEW RETURNED	21-Oct-2019

GENERAL COMMENTS	I have got couple of further comments:
--

	1. I still find it inappropriate to distinguish between epidemiological models (that are based on projections of prevalence) and Markov chain model, and I am not convinced by the argumentation presented by the authors. All the models entertained in this study may be classified with the high-level term 'epidemiological model' 2. P. 7, lines 9-16 (QOF data). It should be made clear already at this point whether the data ('numbers diagnosed with diabetes by general practitioners in England') represent incidence or prevalence data. From subsequent sections it seems that it is prevalence
--	---

REVIEWER	Igor Akushevich Duke University
REVIEW RETURNED	19-Nov-2019

GENERAL COMMENTS	Unfortunately the authors did not respond to reviewer's comments appropriately. The reviewer's comments were written to help the authors to better outline and specify their research in the area of projections of diabetes prevalence. The authors want to restrict the class of considered models by the models that are transparent and simple. This research objective is possible but can be of interest only together with detailed description of the place of this class of models among other models also available in literature. In addition any projection model (especially a simple or/and transparent model usually constructed with multiple assumptions) has important properties which have to be clarified for the readers. I think most important of them are i) populations used for model estimation and ii) properties of transition probabilities, e.g., their age dependence. I believe that authors' response to attempts of reviewers to clarify these points are not sufficient.
--

VERSION 2 – AUTHOR RESPONSE

Professor Anders Green

Comment

1. I still find it inappropriate to distinguish between epidemiological models (that are based on projections of prevalence) and Markov chain model, and I am not convinced by the argumentation presented by the authors. All the models entertained in this study may be classified with the high-level term 'epidemiological model'.

Response

We now appreciate the importance of Professor Green's comment given the new modelling approach developed by Akushevich and his colleagues.

We have changed our description of 'epidemiological models' to 'prevalence based models'.

Comment

2. P. 7, lines 9-16 (QOF data). It should be made clear already at this point whether the data ('numbers diagnosed with diabetes by general practitioners in England') represent incidence or

prevalence data. From subsequent sections it seems that it is prevalence death, but at least this aspect should be discussed appropriately in the paper.

Response

We have revised the text to make this clear:

We estimated, by OLS regression analysis (using R),^[2] the trend increase in the reported prevalence of diabetes as diagnosed by general practitioners in England, in the Quality Outcomes Framework (QOF) from 2004-05 (2004) to 2017-18 (2017)).^[3]

Professor Igor Akushevich

Comment

Professor Akushevich argues that we did not respond appropriately to his earlier comments and observes that:

The authors want to restrict the class of considered models by the models that are transparent and simple. This research objective is possible but can be of interest only together with detailed description of the place of this class of models among other models also available in literature. In addition any projection model (especially a simple or/and transparent model usually constructed with multiple assumptions) has important properties which have to be clarified for the readers. I think most important of them are i) populations used for model estimation and ii) properties of transition probabilities, e.g., their age dependence. I believe that authors' response to attempts of reviewers to clarify these points are not sufficient.

Response to comments by Professor Akushevich

Our critique of the prevalence-based models we reviewed has been developed to highlight the limitations of a focus on prevalence and the importance of being able to understand its drivers of disease incidence and patient survival. And, because prevalence is the difference between those random variables, models that project prevalence will be expected to have large errors in estimation, which emphasises the importance of reporting confidence interval estimates.

The models developed by Professor Akushevich and colleagues, of disease incidence and patient survival, are estimated from a single data set, incorporate changes over time and take account of age. As the Markov chain models we reviewed, also aim to model disease incidence and patient survival, we have considered the issues raised by Professor Akushevich over data and the impact of age on transition probabilities. We point out that the models we reviewed use different data sets from different populations to estimate transitions between states other than death and mortality. And that transition probabilities typically do not change over time, although for seven models they do vary by age.

In our discussion we refer to the recently-published paper by Leal et al.^[4], who report a systematic review of 'Decision Models of prediabetes populations' that reported economic outcomes or evaluations. They found that the majority of models they reviewed 'assumed that the rate of progression to T2D was constant across the entire prediabetes population', which they attribute, in part to limitations in the available data; and that 'few models reported any model validation'. They recommend the development of 'more comprehensive models that are capable of better capturing the continuity in disease progression and, also, of incorporating the identification of novel biomarkers'. They recognise such models require more detailed data and only need to be comprehensive enough

to provide reliable estimates for decision making. We point out that simple models may be useful for economic analyses of preventive interventions, but, they do need to be validated, given the mix of sources of data used and simplifying assumptions made.

We conclude that 'to inform national policies, governments need estimates of the impacts of preventive interventions on reducing the burden of disease from T2D in the general population. These estimates ought to be derived from validated models, designed to use available data, that estimate changes over time in the incidence and survival of patients with T2D, with and without preventive interventions'.

References

- 1 Akushevich I, Yashkin AP, Kravchenko J, et al. Identifying the causes of the changes in the prevalence patterns of diabetes in older U.S. adults: A new trend partitioning approach. *J Diabetes Complications* Published Online First: 2018. doi:10.1016/j.jdiacomp.2017.12.014
- 2 RStudio Team. *RStudio: Integrated Development for R*. 2015.
- 3 NHS Digital. *Quality and Outcomes Framework (QOF) - 2016-17*. 2017.
- 4 Leal J, Morrow LM, Khurshid W, et al. Decision models of prediabetes populations: A systematic review. *Diabetes, Obes Metab* Published Online First: 2019. doi:10.1111/dom.13684